# Hybrid BW-EDAS MCDM methodology for optimal industrial robot selection

Tabasam Rashid[1☯], Asif Ali[1☯], Yu-Ming Chu[2☯¤]*

**1** Department of Mathematics, School of Sciences, University of Management and Technology, Lahore, Pakistan, **2** Hunan Provincial Key Laboratory of Mathematical Modeling and Analysis in Engineering, Changsha University of Science Technology, Changsha, P.R. China

☯ These authors contributed equally to this work.
¤ Current address: Department of Mathematics, Huzhou University, Huzhou, P. R. China
* chuyuming2005@126.com

**Data Availability Statement:** All relevant data are within the manuscript.

**Funding:** Dr. Yu-Ming Chu awarded with the research grant by the National Natural Science Foundation of China (Grant Nos. 11971142,

## Abstract

Industrial robots have different capabilities and specifications according to the required applications. It is becoming difficult to select a suitable robot for specific applications and requirements due to the availability of several types with different specifications of robots in the market. Best-worst method is a useful, highly consistent and reliable method to derive weights of criteria and it is worthy to integrate it with the evaluation based on distance from average solution (EDAS) method that is more applicable and needs fewer number of calculations as compared to other methods. An example is presented to show the validity and usability of the proposed methodology. Comparison of ranking results matches with the well-known distance-based approach, technique for order preference by similarity to ideal solution and VIseKriterijumska Optimizacija I Kompromisno Resenje (VIKOR) methods showing the robustness of the best-worst EDAS hybrid method. Sensitivity analysis performed using eighty to one ratio shows that the proposed hybrid MCDM methodology is more stable and reliable.

## Introduction

Massive utilization of robots in industries is due to extensive progress in engineering and information technology. Robots have so many features, specifications and capabilities to do work accurately as compared to manpower. A robot is a multipurpose machine that is self-controlled and reprogrammable, and can perform various tasks in diverse industrial applications for example welding, spray painting, loading, finishing, assembly, etc. [1–3]. Utilization of robots enhanced the productivity and profitability of organizations. Factors like operation speed, quality, reliable production process, etc. are enhanced by the utilization and implementation of modern technology in the organizations. Further, due to the vast competitive market, it becomes very difficult for companies and organizations to choose proper machines/robots that best fit their requirements. The key point in the selection process of a robot is to identify the attributes according to the requirement of the work [4]. These attributes are classified into two categories, benefit and non benefit, benefit attributes need to be high in value and on

61673169) http://www.nsfc.gov.cn/english/site_1/index.html. The funders had no role in study design, data collection and analysis, decision to publish, or preparation of the manuscript.

**Competing interests:** The authors have declared that no competing interests exist.

contrary non benefit should have low values. e.g. cost is non benefit so require least value and repeatability is benefit attribute so require highest value. Table 1 represents the acronyms with their descriptions that are used in this paper.

There are many MCDM methodologies for suitable industrial robot selection. Knott and Getto [5] developed a robot selection methodology considering uncertainty of prediction time, labour components, net present value of several alternatives evaluated at the same time reference and overheads of alternatives. Dooner [6] simulated robot operations in workspace and workspace is aided for robot selection. According to Hinson [7] work environment is the major factor for the selection of robot. Huang and Ghandforoush [8] evaluated robots based on budget requirement and investment etc. Jones et al. [9] pointed out the importance of marginal value function for the selection of robots. Imany and Schlesinger [10] proposed a linear goal programming model for robot selection and presented a comparative analysis with ordinary least square methods. A fuzzy method is applied by Wang et al. [11] in their decision support system for robot selection. Boubekri et al. developed an expert system for the selection of industrial robot considering functional, economic and organizational factors. Agrawal et al. [12] proposed robots selection methodology based on TOPSIS and presented a decision support system to ease inexperienced users, the drawback of the method is not considering the qualitative nature of attributes. Booth et al. [13] considered both Mahalanobis distance and principal components analysis for their robot selection model. Liang and Wang [14] proposed robot selection algorithm by integrating fuzzy sets with hierarchical structure and made preferences using fuzzy suitability index. Khouja and Offodile [15] tried to highlight the future research directions for the selection problem of industrial robot by comprehensive literature review. Khouja [16] presented robot selection two-phase model, in first phase DEA is applied, and the second phase is a multiple attribute decision-making model. Goh et al. [17] presented a revised weighted sum decision model for ranking robots selection. Parkan and Wu [18] proposed operational competitiveness rating process and compared it with some existing multiple criteria decision making robot selection methods. Chu and Lin

**Table 1. Description of used acronyms.**

| Acronym | Description |
| --- | --- |
| AHP | Analytical Hierarchy Process |
| FAHP | Fuzzy AHP |
| AS | Appraisal Score |
| BWM | Best-Worst Method |
| DEA | Data development Analysis |
| DBA | Distance-Based Approach |
| EDAS | Evaluation Based on Distance from Average Solution |
| ELECTRE | ELimination and Et Choice Translating REality |
| IVIHF | Interval-Valued Intuitionistic Hesitant Fuzzy |
| MCDM | Multi Criteria Decision Making |
| NDA | Negative Distance from Average |
| PDA | Positive Distance from Average |
| SWARA | Step-wise Weight Assessment Ratio Analysis |
| TOPSIS | Technique for Order of Preference by Similarity to Ideal Solution |
| FTOPSIS | Fuzzy TOPSIS |
| TPOP | Technique of Precise Order Preferences |
| VIKOR | VIseKriterijumska Optimizacija i kompromisno Resenje |
| FVIKOR | Fuzzy VIKOR |

[19] proposed FTOPSIS method for industrial robot selection by pointing out the limitations of the Liang and Wang [14] method in which they violated the basic fuzzy logic rules to convert objective values of attributes into fuzzy values. Bhangale et al. [20] ordered a large number of robot selection criteria with the help of graphical and TOPSIS methods, his weight assigning method was not consistent. Rao and Padmanabhan [1] proposed the diagraph and matrix approach, but this becomes complex for a large number of robot selection attributes. Kahraman et al. [21] considered economical and technical criteria for industrial robot selection with proposed fuzzy hierarchical TOPSIS model. Karsak [22] proposed an integrated robot selection methodology based on quality function deployment and fuzzy linear regression. Chatterjee et al. [2] compared relative performance of an industrial application using outranking methods, VIKOR and ELECTRE but they require more computations. Kumar and Garg [23] proposed a quantitative DBA method for robot selection, this method has a limitation of not considering the qualitative nature of attributes. Tansel, Yurdakul, and Dengiz [24] presented ROBSEL (a two-phase decision support system for robot selection) method with FAHP that reduces the expert dependency for the robot selection process. Bairagi, Dey, Sarkar, and Sanyal [25] used FAHP to assign weights to the criteria and applied FTOPSIS, FVIKOR and complex proportional assessment of alternatives with grey relations for robot ranking. Liu et al. [26] used interval 2-type linguistic fuzzy set for criteria weights and used these weight values in TOPSIS method for robot selection. Rashid et al. [27] proposed generalized interval-valued trapezoidal FTOPSIS method for robot selection using linguistic terms. Parameshwaran et al. [28] calculated weights using FAHP and then ranking of educational purpose robot selection is done using FTOPSIS, FVIKOR and fuzzy Delphi methods. Bairagi et al. [29] proposed TPOP that improved inconsistency in preference order of alternatives by using advanced version of entropy to calculate weights and done comparative analysis. Ghorabaee [30] proposed FVIKOR industrial robot selection method using interval type-2 fuzzy sets, and analyzed the stability of the proposed method using the Spearman correlation coefficient and comparative study with existing methods. Joshi and Kumar [31] extended the TOPSIS method and introduced Choquet integral operator for IVIHF set for the appropriate robot selection. Narayanamoorthy et al. [32] proposed IVIHF entropy method and used FVIKOR techniques to make priority of industrial robot. Ali and Rashid [33] proposed group BWM for industrial robot selection by weighting decision makers based on their previous experience by an executive and showed the robustness of the proposed method by comparative study and checking minimum violation and total deviation. All these methods have their advantages and disadvantages, most methods, especially fuzzy methods, have extensive calculation work.

Rezaei [34, 35] presented BWM in 2015 that is a refined form of AHP method with more consistency and less pairwise comparisons. Rezaei et al. [36, 37] applied BWM to link supplier development with the supplier segmentation. Gupta and Barua [38] used BWM to rank enablers of technological innovation by identifying them. Gupta et al. [39] ranked barriers to energy efficiency in building by the utilization of BWM to the development and improvement of energy efficiency measures. Gupta [40] developed hybrid best-worst VIKOR method for prioritization of service quality attributes to evaluate and enhance service quality of the airline industry and evaluated ranking of best airline. Ren [41] employed BWM for the technology selection for ballast water treatment and determined their grades. Ren et al. [42] solved urban sewage sludge treatment problem with the help of BWM. Ali Torabi et al. [43] applied BWM for a business continuity management system which is an enhanced risk assessment framework. Kaa et al. [44] applied BWM to evaluate the competitive advantage of fuel cell and battery-powered electric vehicles. Rezaei et al. [45] solved a supplier management problem with the help of BWM by proposing purchasing portfolio matrix hybrid with

supplier potential matrix. The supply chain is a sensitive problem for the production industry that is solved by many researchers with the help of BWM, that is a great success of the method [37, 45–51]. Parkash Garg and Sharma [52] proposed VIKOR method hybrid with BWM for the selection and evaluation of sustainable outpouring partner. Mokhtarzadeh et al. [53] prioritize twenty-three technology options for a high-tech company by finding their weights with the help of BWM (a case study in Iran). Zolfani and Chatterjee [54] proposed a comparative study of BWM and SWARA methods for the household furnishing materials selection. Serrai et al. [55] evaluated web service selection using BWM and compared it with the results of VIKOR, simple additive weighting, TOPSIS and complex proportional assessment methods. Uncertain extensions of BWM are proposed by different researchers [56–60] these extensions have different advantages but require extensive calculations. Pamucar et al. [61] proposed a new full consistency method (FUCOM) for criteria weight calculations showing the method perform better then the BWM and AHP method with respect to consistency and pairwise comparisons but the method require an initial priority of criteria by the decision-maker or expert on the basis of their experience or preference that can confuse DMs to make proper preferences but using BWM DMs require only to select most favorable and least favorable criteria and make best to other and others to worst pairwise comparisons that is much easy task. However, FUCOM is also being integrated with other MCDM methods for selection problems [62, 63].

Ghorabaee et al. [64] proposed EDAS a multiple criteria decision-making method in 2015 and used it for the classification of inventory. EDAS is a compensatory method in which criteria are independent, for evaluation by EDAS qualitative attributes are converted to quantitative, decision matrix determines the input information and using this method excavator is selected for a road construction company [65]. Aggarwal et al. [66] applied EDAS method for the selection of an appropriate smart-phone within the particular budget, particularly in the Indian market. Kundakci [67] applied EDAS method combined with measuring attractiveness by a categorical based evaluation technique for the selection of steam boiler for dyehouse of a textile company. Ecer [68] applied EDAS along with FAHP and Delphi technique for group decision selection of the best among four third-party logistics services for a marble company. Stevic et al. [69] applied AHP and EDAS method for the selection of one of the four scenarios are evaluated. EDAS is prominent as its solution is obtained from the average solution that eliminates the unfairness risk of the experts towards an alternative. Simplicity and need for fewer computations are the most significant characteristics of the EDAS method. BWM got a huge success in application are due to its consistency and fewer pairwise comparisons. Similarly, application of EDAS method is also very vast due to its simplicity and robustness. So, it is advantageous to integrate BWM with the EDAS method as BWM is more applicable and more consistent for weight calculation and EDAS method provide more stable results with low calculation cost.

In this paper, a hybrid best-worst EDAS method is proposed for industrial robot selection. The motivation of this paper is to provide a simple, reliable, and robust MCDM methodology for industrial robot selection with fewer calculation cost. No one integrated BWM with EDAS method for robot selection problem. There are three advantages of BWM method for weight calculations, 1) BWM provide consistent results, 2) it required fewer pairwise comparisons as compared to other MCDM methods, 3) selecting the best and the worst criteria and comparing with other criteria is much easy for DMs using 1 to 9 scale. EDAS method is selected for ranking the robots as it is new method and have wide application area, it requires low calculation cost as compared to other MCDM methods. The ranking results are compared with TOPSIS, VIKOR and DBA methods. Sensitivity analysis performed with respect to criteria shows that $s_3$ and $s_4$ are sensitive to assign weighs and are more important for selection process.

## Best-worst method

AHP is a more applicable and frequently used method but has drawbacks of consistency and need for more comparisons [34]. Rezaei remedy these issues by presenting BWM in 2015 [34]. BWM is a pairwise comparison weight deriving process that is more consistent, need fewer pairwise comparisons and hence is more reliable. BWM comprises of five steps for calculating the weights of criteria.

**Step 1:** First step involves to select decision criteria sets.

**Step 2:** In this step, decision makers decide for the best or more favorable criterion and the worst or least favorable criterion e.g. load capacity may be best criterion and vendor's service may be worst criterion.

**Step 3:** In this step, preference of the best criterion over all the other criteria are determined using 1 to 9 scale represented by $A_{BO} = (s_{B1}, s_{B2}, \ldots, s_{Bn})$, where $s_{Bj}$ represent the preference of the best criterion $B$ over the criteria $j$.

**Step 4:** Preference of all the other criteria over the worst criterion is determined using 1 to 9 scale by the decision makers and is represented by $A_{OW} = (s_{1W}, s_{2W}, \ldots, s_{nW})$, where $s_{jW}$ represent the preference of all the criteria $j$ over the worst criterion $W$.

**Step 5:** Fifth step is to determine weights $(w_1^*, w_2^*, ..., w_n^*)$ of criteria.

BWM have three advantages 1) it is always consistent, 2) it require fewer comparisons as compared to AHP, 3) it require fewer calculations. Eq (1) represent the mathematical model of BWM.

$$\left. \begin{array}{l} \min \ \max\left\{ \left| \dfrac{w_B}{w_i} - s_{Bi} \right|, \left| \dfrac{w_i}{w_W} - s_{iW} \right| \right\} \\ s.t. \\ \sum_i w_i = 1 \\ w_i \geqslant 0, \ \text{for all} \ i \end{array} \right\} \tag{1}$$

The equations in (1) can easily be converted in the form represented by the equations in (2).

$$\left. \begin{array}{l} \min \ \varepsilon \\ s.t. \\ \left| \dfrac{w_B}{w_i} - s_{Bi} \right| \leq \varepsilon \\ \left| \dfrac{w_i}{w_W} - s_{iW} \right| \leq \varepsilon \\ \sum_i w_i = 1 \\ w_i \geq 0 \quad \text{for all} \ i \end{array} \right\} \tag{2}$$

Solution of the equations in (2) are the weights of criteria $(w_1^*, w_2^*, w_3^*, \ldots, w_n^*)$. Using BWM reliability of comparisons is determined using consistency ratio (CR) that is calculated using

**Table 2. The CI values for the comparisons using BWM.**

| $s_{BW}$ | 1 | 2 | 3 | 4 | 5 | 6 | 7 | 8 | 9 |
|---|---|---|---|---|---|---|---|---|---|
| CI | 0.00 | 0.4384 | 1.00 | 1.6277 | 2.2984 | 3.00 | 3.7251 | 4.4689 | 5.2280 |

Eq (3), $\varepsilon^*$ (optimal value obtained by the solution of the model (2) and its consistency index (CI) [34] whose values are taken from Table 2.

$$CR = \frac{\varepsilon^*}{CI} \tag{3}$$

## EDAS method

In EDAS method, PDA solution and NDA solution are calculated, the optimal alternative has the higher distance from the nadir solution and lowest distance from the ideal solution. This method is useful for conflicting criteria and worthy due to need for fewer calculations. EDAS method comprises the following steps.

**Step 1:** First step involved the selection of the most important criteria for the alternatives.

**Step 2:** Construction of the decision-matrix ($M$), presented by Eq (4):

$$M = [M_{ij}]_{r \times k} = \begin{bmatrix} m_{11} & m_{12} & \ldots & m_{1k} \\ m_{21} & m_{22} & \ldots & m_{2k} \\ \vdots & \vdots & \vdots & \vdots \\ m_{r1} & m_{r2} & \ldots & m_{rk} \end{bmatrix} \tag{4}$$

where $m_{ij}$ determines the performance value of $i$th alternative with respect to $j$th criterion.

**Step 3:** In this step, Eq (5) is used to determine the average solution to all criteria:

$$AV = [AV_j]_{1 \times k} \tag{5}$$

where, $AV_j = \frac{\sum_{i=1}^{r} m_{ij}}{r}$.

**Step 4:** The matrix PDA and NDA are calculated according to the benefit and cost criteria as follows:

$$PDA = [PDA_{ij}]_{r \times k}$$

$$NDA = [NDA_{ij}]_{r \times k}$$

if $j$th criterion is beneficial,

$$PDA_{ij} = \frac{max(0, (m_{ij} - AV_j))}{AV_j}$$

$$NDA_{ij} = \frac{max(0, (AV_j - m_{ij}))}{AV_j}$$

and if $j$th criterion is non-beneficial,

$$PDA_{ij} = \frac{max(0, (AV_j - m_{ij}))}{AV_j}$$

$$NDA_{ij} = \frac{max(0, (m_{ij} - AV_j)))}{AV_j}$$

where $PDA_{ij}$ and $NDA_{ij}$ denote the positive distance and the negative distance of $i$th alternative from average solution in terms of $j$th criterion, respectively.

**Step 5:** The weighted sum of PDA and NDA are determined at this step and are represented by the Eqs (6) and (7):

$$SP_i = \sum_{j=1}^{k} w_j PDA_{ij} \tag{6}$$

$$SN_i = \sum_{j=1}^{k} w_j \mathbf{NDA}_{ij} \tag{7}$$

where $w_j$ represents the weight of $j$th criterion.

**Step 6:** The values of $SP$ and $SN$ are normalize, shown as follows:

$$NSP_i = \frac{SP_i}{max_i(SP_i)}$$

$$NSN_i = \frac{SN_i}{max_i(SN_i)}$$

**Step 7:** The appraisal score ($AS$) are calculated for all the alternatives by using Eq (8)):

$$AS_i = \frac{1}{2}(NSP_i + NSN_i) \tag{8}$$

Best-worst method is integrated with EDAS method. Weights derived using BWM are used to make priority ranking of robots using EDAS method. The proposed hybrid multiple criteria decision making methodology is represented in Fig 1.

## Ranking evaluation for robot selection

In this section, weights are derived using BWM, and then EDAS method evaluated the ranking for the best selection of robot using these weights. Load capacity $s_1$, repeatability $s_2$, velocity ratio $s_3$ and degree of freedom $s_4$ are considered as criteria for industrial robot selection, here decision makers determines $s_3$ as the worst criterion and $s_2$ as the best criterion. Table 3 represents the comparisons of the criteria by the decision-makers.

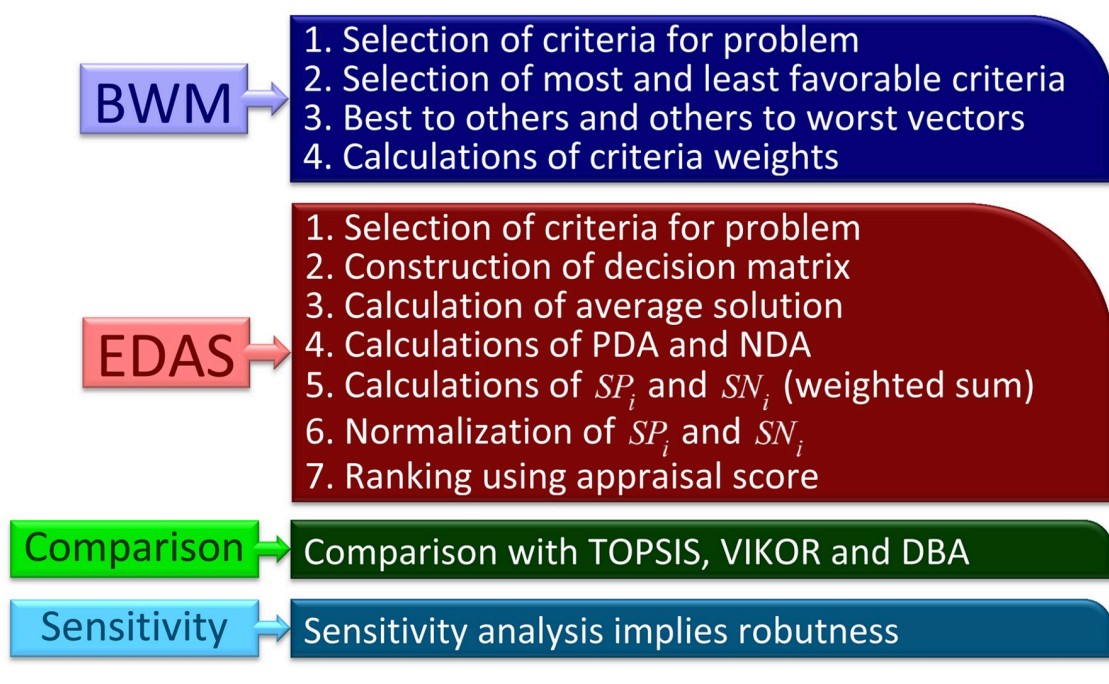

**Fig 1. Proposed hybrid MCDM methodology.** BWM: Best-worst method. EDAS: Evaluation based on distance from average solution.

The weights of criteria are evaluated using model (9).

$$
\left.
\begin{aligned}
&\min \varepsilon \\
&s.t. \\
&\left|\frac{w_2}{w_1} - 7\right| \le \varepsilon \quad ; \quad \left|\frac{w_2}{w_3} - 8\right| \le \varepsilon \\
&\left|\frac{w_2}{w_4} - 4\right| \le \varepsilon \quad ; \quad \left|\frac{w_1}{w_3} - 3\right| \le \varepsilon \\
&\left|\frac{w_4}{w_3} - 6\right| \le \varepsilon \\
&w_1 + w_2 + w_3 + w_4 = 1 \\
&w_i \ge 0 \quad \text{for} \quad i = 1, 2, 3, 4.
\end{aligned}
\right\}
\tag{9}
$$

The solution of the model (9) will provide interval solutions for each $w_i$, average to which will provide the weights of criteria. i.e. $w_1^* = 0.0939$, $w_2^* = 0.5875$, $w_3^* = 0.0604$ and $w_4^* = 0.2582$. Where $\xi^* = 1.7251$, $CI = 4.47$ and $CR = 0.3859$.

**Table 3. Best to others and others to worst criteria comparisons.**

|  | $s_{21}$ | $s_{23}$ | $s_{24}$ | $s_{13}$ | $s_{43}$ |
|---|---|---|---|---|---|
| Senior Expert | 7 | 8 | 4 | 3 | 6 |

**Table 4. Evaluation of alternatives with respect to criteria [20].**

| Alternatives | $s_1$ | $s_2$ | $s_3$ | $s_4$ |
|---|---|---|---|---|
| Robot 1 | 60 | 0.40 | 125 | 5 |
| Robot 2 | 60 | 0.40 | 125 | 6 |
| Robot 3 | 68 | 0.13 | 75 | 6 |
| Robot 4 | 50 | 1.00 | 100 | 6 |
| Robot 5 | 30 | 0.60 | 55 | 5 |

Table 4 represents the evaluation of robot 1, robot 2, robot 3, robot 4 and robot 5 with respect to the load capacity, repeatability, velocity ratio, and degree of freedom.

The matrices (10) and (11) represents the positive distance from average and the negative distance from average, respectively. Calculations of $SP$, $NSP$, $SN$, $NSN$, $AS$ and ranking are determined in Table 5.

$$PDA = \begin{pmatrix} 0.1194 & 0.2095 & 0.3021 & 0 \\ 0.1194 & 0.2095 & 0.3021 & 0.0714 \\ 0.2687 & 0.7431 & 0 & 0.0714 \\ 0 & 0 & 0.0417 & 0.0714 \\ 0 & 0 & 0 & 0 \end{pmatrix} \quad (10)$$

$$NDA = \begin{pmatrix} 0 & 0 & 0 & 0.1071 \\ 0 & 0 & 0 & 0 \\ 0 & 0 & 0.2188 & 0 \\ 0.0672 & 0.9763 & 0 & 0 \\ 0.4403 & 0.1858 & 0.4271 & 0.1071 \end{pmatrix} \quad (11)$$

## Comparison of results

The final results of EDAS method are compared with DBA, TOPSIS and VIKOR methods represented in Table 6 and noted that the ranking results for all these methods are the same for the weights derived by the BWM.

Graphical comparison of normalized ranking score values of EDAS and TOPSIS methods are expressed in Fig 2 where alternatives are ranked according to the decreasing score values

**Table 5. Values of $SP$, $NSP$, $SN$, $NSN$, $AS$ and ranking for the EDAS method.**

| $i$ | $SP_i$ | $NSP_i$ | $SN_i$ | $NSN_i$ | $AS_i$ | Ranking |
|---|---|---|---|---|---|---|
| 1 | 0.1525 | 0.3176 | 0.0277 | 0.9523 | 0.6350 | 3 |
| 2 | 0.1710 | 0.3560 | 0.0000 | 1.0000 | 0.6780 | 2 |
| 3 | 0.4802 | 1.0000 | 0.0132 | 0.9772 | 0.9886 | 1 |
| 4 | 0.0210 | 0.0437 | 0.5798 | 0.0000 | 0.0218 | 5 |
| 5 | 0.0000 | 0.0000 | 0.2039 | 0.6483 | 0.3241 | 4 |

**Table 6. Comparison of TOPSIS, VIKOR and DBA methods with proposed method.**

| Alternatives | Proposed Method | Rank | TOPSIS | Rank | VIKOR | Rank | DBA | Rank |
|---|---|---|---|---|---|---|---|---|
| Robot 1 | 0.6350 | 3 | 0.6877 | 3 | 0.5013 | 3 | 0.7636 | 3 |
| Robot 2 | 0.6780 | 2 | 0.6913 | 2 | 0.2436 | 2 | 0.5524 | 2 |
| Robot 3 | 0.9986 | 1 | 0.9668 | 1 | 0.0000 | 1 | 0.1093 | 1 |
| Robot 4 | 0.2183 | 5 | 0.0673 | 5 | 0.9444 | 5 | 1.7759 | 5 |
| Robot 5 | 0.3141 | 4 | 0.4554 | 4 | 0.7519 | 4 | 1.1358 | 4 |

and comparison of normalized ranking score values of DBA and VIKOR methods are represented in Fig 3 where alternatives are ranked according to the increasing score values.

## Sensitivity analysis

The sensitivity analysis is the process/tool to check the priority ranking consistency of the MCDM method. The sensitivity analysis is done with assigning eighty per cent of total weight to one criterion and the rest of the weight to all the criteria with equal strength, methodology is adapted from the Jain et al. [70]. Different scenarios of the weight selection for criteria is presented in Table 7. Normalized ASs of scenario 1,2,3 and 4 are graphically shown in Figs 4–7, respectively, and its corresponding ranking effects on alternative robots are shown in Table 8 and Figs 8 and 9. The results show that for scenario 1 and scenario 2 the ranking order of robot 4, robot 5 interchange their position but first three preference do not effect the results. But scenario 3 and scenario 4 are more sensitive to the ranking where Robot 3 that was at first rank gone to the rank 4 but for scenario 3 good thing is that second and third preferences become first and second preferences, respectively but for scenario 4 4th preference becomes 5th preference and fifth becomes first preference that is extremely sensitive result as compared to the proposed result. The results of sensitivity analysis show that criteria $s_3$ and $s_4$ are more sensitivity with respect to assigning weights because they provide more rank reversal for alternative robots as compared to criteria $s_1$ and $s_2$ whose 4th and 5th preferences are sensitive. If we conclude this analysis the criteria $s_3$ and $s_4$ are important criteria for this selections process.

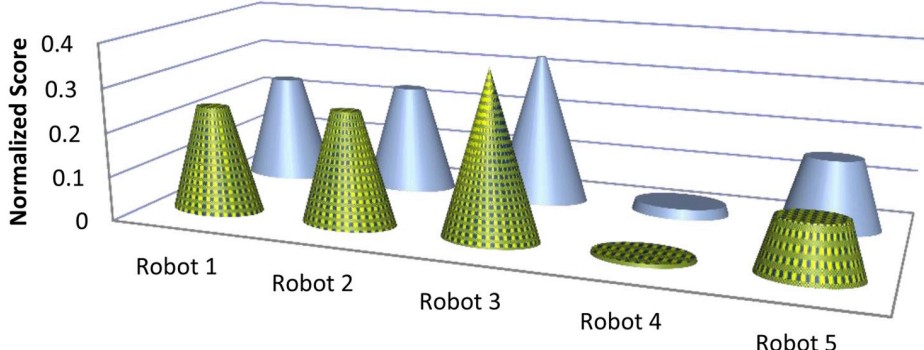

|  | Robot 1 | Robot 2 | Robot 3 | Robot 4 | Robot 5 |
|---|---|---|---|---|---|
| EDAS Results | 0.2398 | 0.2561 | 0.3734 | 0.0082 | 0.1224 |
| TOPSIS Results | 0.2397 | 0.241 | 0.3371 | 0.0235 | 0.1588 |

**Fig 2. Comparison of normalized score of EDAS and TOPSIS methods.**

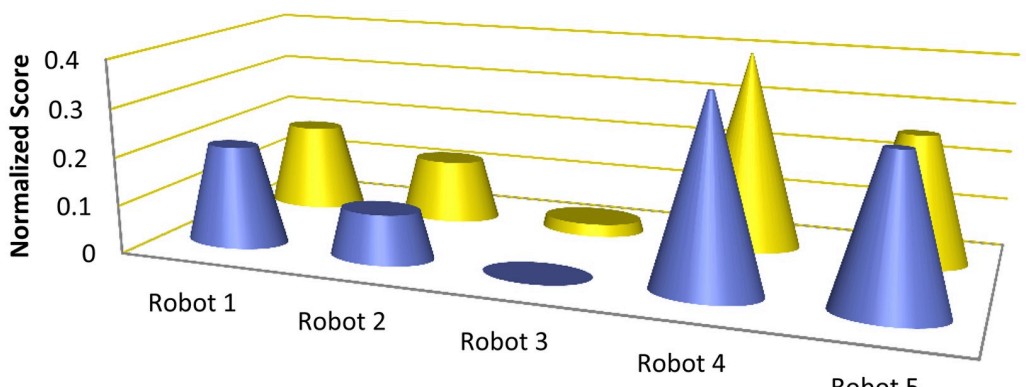

| | Robot 1 | Robot 2 | Robot 3 | Robot 4 | Robot 5 |
|---|---|---|---|---|---|
| ■ VIKOR Results | 0.2054 | 0.0998 | 0 | 0.3869 | 0.308 |
| ■ DBA Results | 0.1761 | 0.1274 | 0.0252 | 0.4095 | 0.2619 |

**Fig 3. Comparison of normalized score of DBA and VIKOR methods.**

**Table 7. Different scenarios for criteria weights.**

| Criteria | Scenario 1 | Scenario 2 | Scenario 3 | Scenario 4 |
|---|---|---|---|---|
| $s_1$ | 0.8000 | 0.0667 | 0.0667 | 0.0667 |
| $s_2$ | 0.0667 | 0.8000 | 0.0667 | 0.0667 |
| $s_3$ | 0.0667 | 0.0667 | 0.8000 | 0.0667 |
| $s_4$ | 0.0667 | 0.0667 | 0.0667 | 0.8000 |

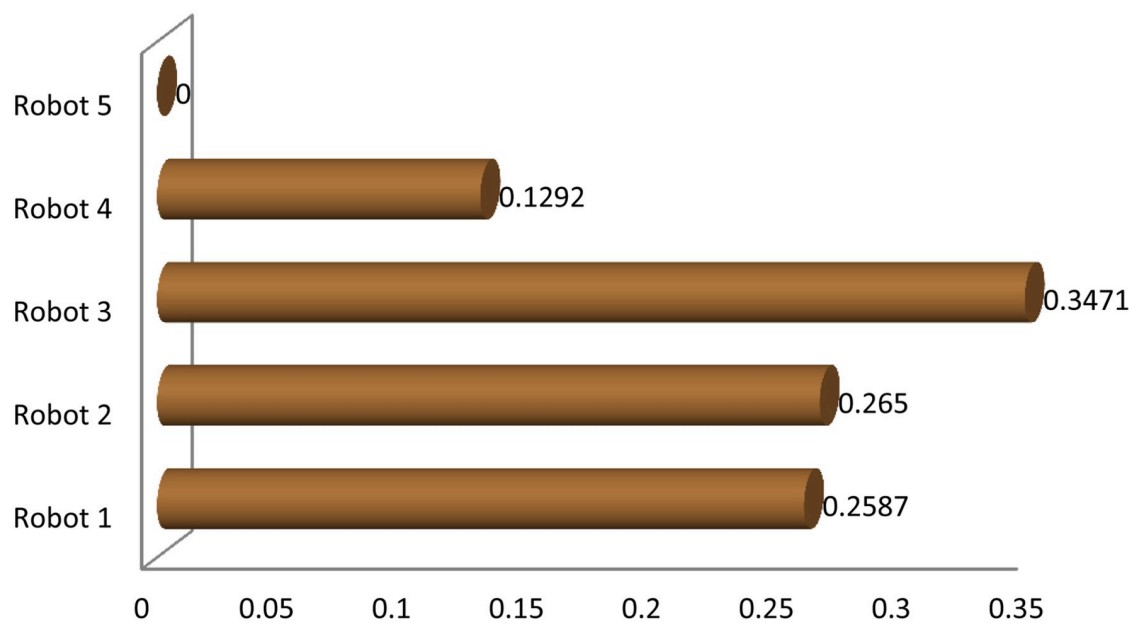

**Fig 4. Sensitivity analysis of the criteria $C_1$.**

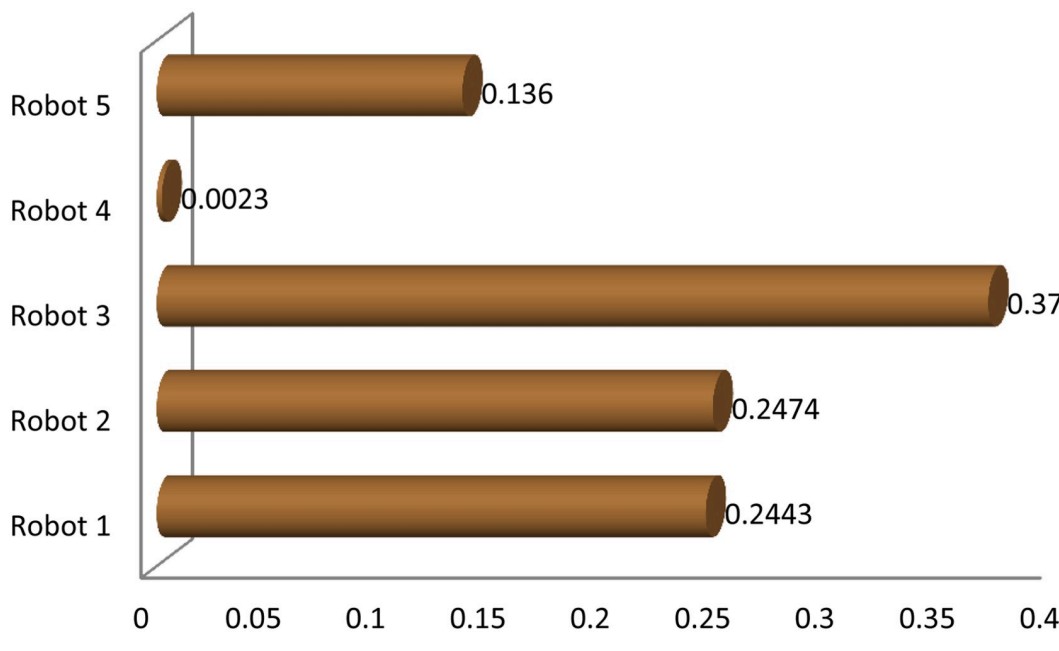

**Fig 5. Sensitivity analysis of the criteria $C_2$.**

## Conclusion

The main reason of the enhanced utilization of robots in the industrial latest manufacturing system is the rapid advancement in the information technology and engineering sciences. Manufacturers, in industrial applications, preferably use robots to perform different repetitious, uncertain and difficult tasks with precision. Hence, for a particular task, to boost the quality of products and enhance productivity in a manufacturing company the most difficult and crucial concern is the selection of proper and suitable robot. To deal with the decision making process, load capacity, repeatability, velocity ratio and degree of freedom are

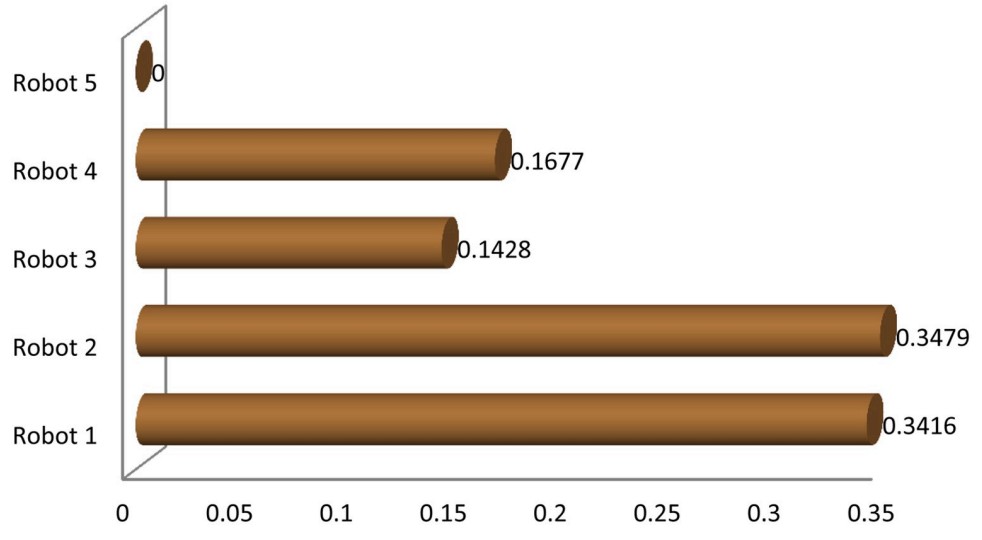

**Fig 6. Sensitivity analysis of the criteria $C_3$.**

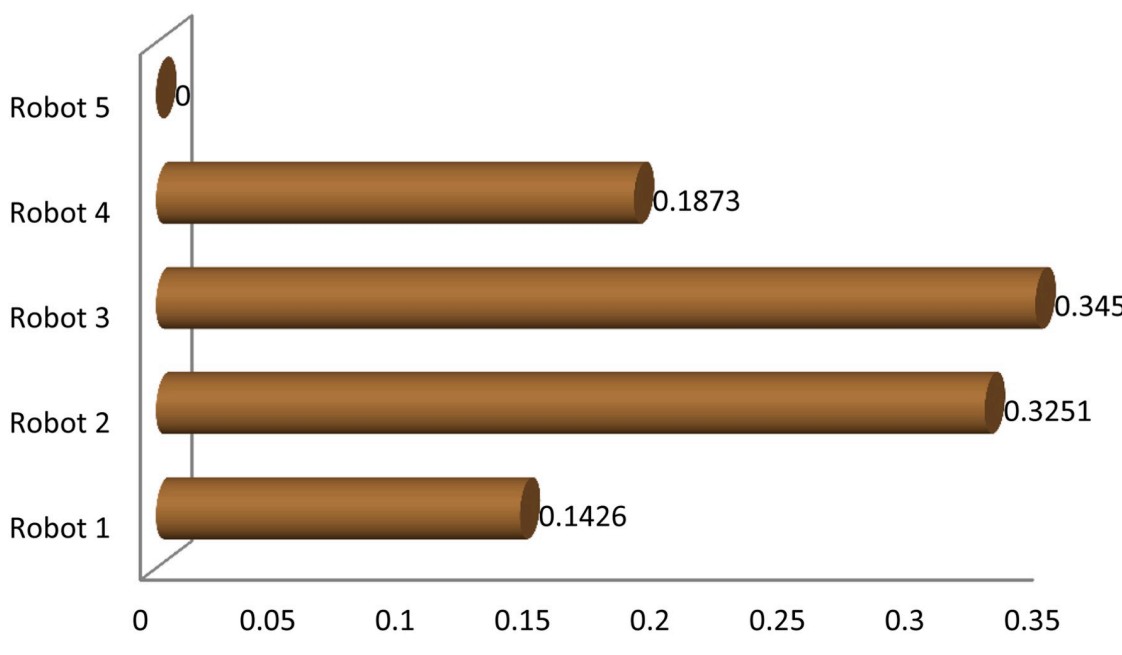

**Fig 7. Sensitivity analysis of the criteria $C_4$.**

considered for appropriate robot selection in industries using the best-worst method integrated with EDAS method. The advantage of the proposed method is that it is more consistent and need fewer calculations. The proposed methodology is more reliable and robust as its ranking results match with the well known existing methods. Sensitivity analysis shows that the results are stable for criteria $s_1$ and $s_2$ and sensitive with respect to $s_3$ and $s_4$. This methodology have following advantages, 1) weight deriving process is consistent, 2) it required fewer pairwise comparisons, 3) user friendly for DMs to provide opinions, 4) less calculation cost. The proposed hybrid BW-EDAS methodology can be used for any number of criteria, qualitative or quantitative, to make preference ranking of the robots. The proposed methodology is a general procedure that can help decision makers to solve any industrial selection problem having finite selection criteria. In future, we will utilize FUCOM method to draw weights and will use EDAS methods for ranking process and conduct a comparative analysis with our proposed method. The work can also be extended in fuzzy environment.

## Managerial applications

A hybrid decision methodology has been developed to evaluate the optimal robot for industrial application. The firms can derive advantage from the decision methodology developed in the study, which can be employed as a road map to a consensus understanding to assess firms'

**Table 8. Ranking of robots with respect to different scenarios defined in Table 7.**

| Alternatives | $s_1$ | $s_2$ | $s_3$ | $s_4$ |
|---|---|---|---|---|
| Robot 1 | 3 | 3 | 2 | 4 |
| Robot 2 | 2 | 2 | 1 | 2 |
| Robot 3 | 1 | 1 | 4 | 1 |
| Robot 4 | 4 | 5 | 3 | 3 |
| Robot 5 | 5 | 4 | 5 | 5 |

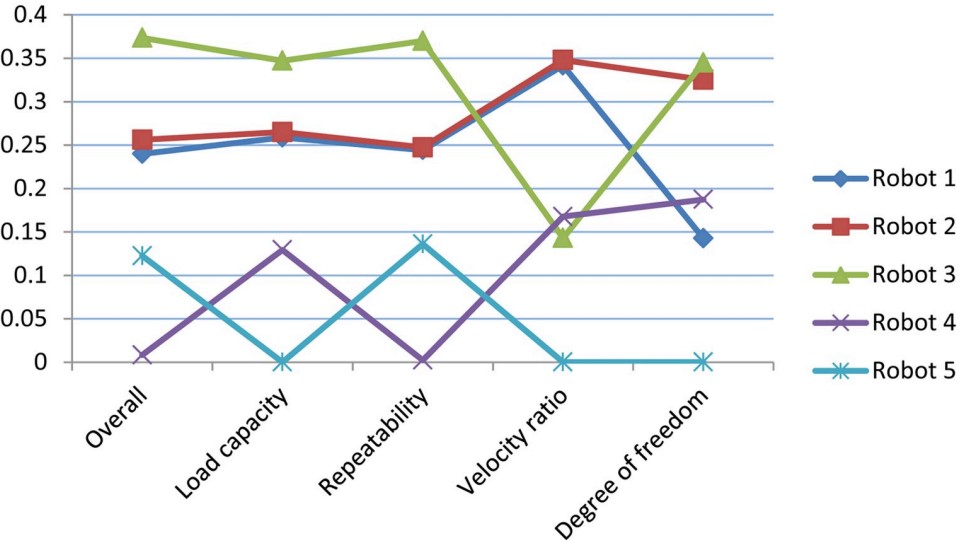

**Fig 8. Sensitivity analysis diagram of the EDAS results.**

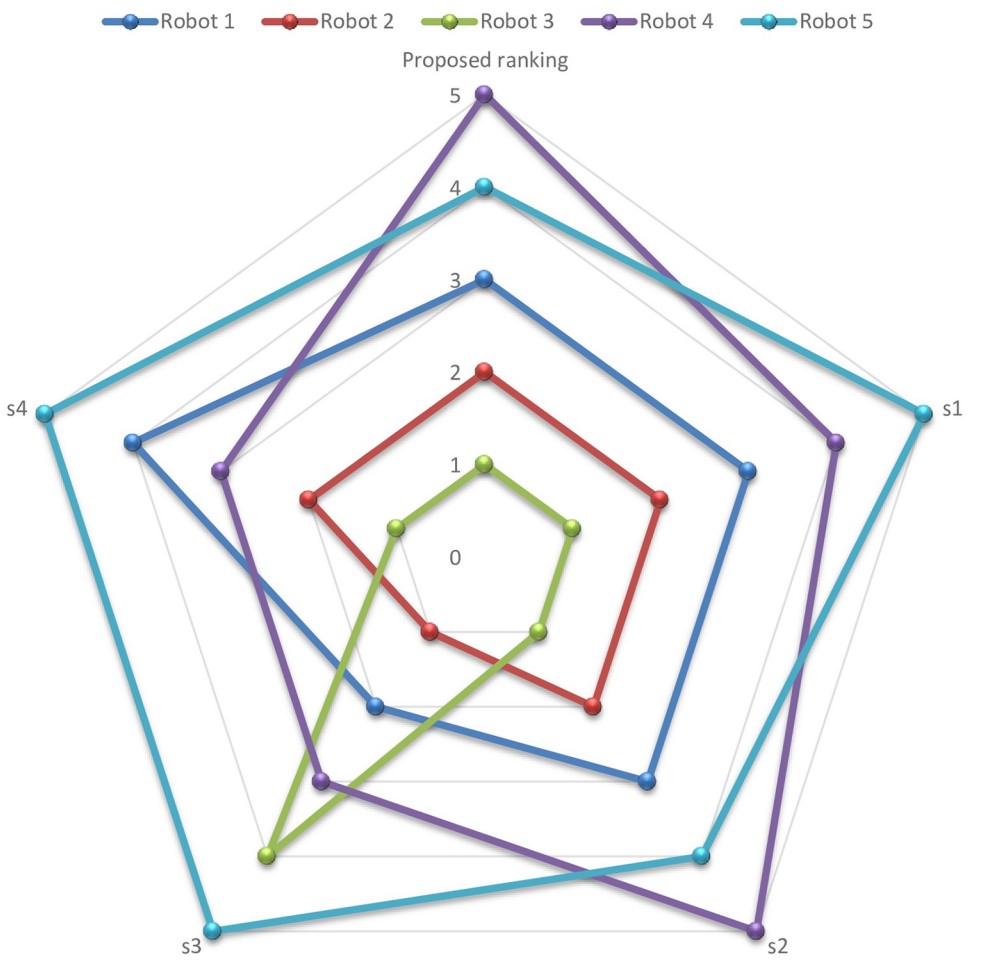

**Fig 9. Sensitivity analysis of robots ranking.**

activities robot selection. Based on the findings of this study, with the optimal robot evaluation tool developed, now firms can determine the various ways to enhance their production and quality assurance. Thus, managers can develop a resilient relationship with their partners, depending on their strengths and take necessary actions to overcome the weaknesses. The digitalization of the firms can be possible by adopting Industry 4.0 approaches and sustainability-related issues for systematically analyzing the decision problem. Moreover, the results of the study can also assist managers in selecting the best robot for specific requirement. Therefore, the results of the study are very significant for implementing the developed decision framework for best fit industrial robot.

## Acknowledgments

We are very thankful to the editor and the anonymous reviewers for their valuable comments and suggestions to improvement the paper.

## Author Contributions

**Formal analysis:** Asif Ali.

**Funding acquisition:** Yu-Ming Chu.

**Investigation:** Asif Ali.

**Methodology:** Tabasam Rashid, Asif Ali.

**Project administration:** Yu-Ming Chu.

**Supervision:** Tabasam Rashid.

**Validation:** Tabasam Rashid.

**Writing – original draft:** Asif Ali.

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
