## [Decision Letter · Decision Letter 0]

1 Jan 2021

PONE-D-20-36145

Hybrid BW-EDAS MCDM Methodology for Optimal Industrial Robot Selection

PLOS ONE

Dear Dr. Chu,

Thank you for submitting your manuscript to PLOS ONE. After careful consideration, we feel that it has merit but does not fully meet PLOS ONE’s publication criteria as it currently stands. Therefore, we invite you to submit a revised version of the manuscript that addresses the points raised during the review process.

We look forward to receiving your revised manuscript.

Kind regards,

Dragan Pamucar

Academic Editor

PLOS ONE

Journal Requirements:

Reviewers' comments:

Reviewer's Responses to Questions

**Comments to the Author**

1. Is the manuscript technically sound, and do the data support the conclusions?

Reviewer #1: Yes

Reviewer #2: Yes

2. Has the statistical analysis been performed appropriately and rigorously? 

Reviewer #1: Yes

Reviewer #2: Yes

3. Have the authors made all data underlying the findings in their manuscript fully available?

Reviewer #1: Yes

Reviewer #2: Yes

4. Is the manuscript presented in an intelligible fashion and written in standard English?

Reviewer #1: Yes

Reviewer #2: Yes

5. Review Comments to the Author

Reviewer #1: Thank you for inviting me as a reviewer for manuscript titled Hybrid BW-EDAS MCDM Methodology for Optimal Industrial Robot Selection; Journal PLOS ONE, Manuscript No: PONE-D-20-36145.

Presented methodology has potential in decision making field and I am giving support to the authors for investigation this topic. The strengths of this paper are: Relevant topic; Flow of the paper; and Explanation of the methods. However, the author(s) need to consider the following points as limitation or further scope for refining the paper:

Need to better highlight the novelty of study in the introduction.

I suggest authors to clearly summarize what specific advantages brings your approach.

The authors didn’t explain why we need this research? What are motivations for this research? Clearly define motivations for your research.

What are limitations of existing approaches in the literature and can be eliminated by this study?

What are benefits of this approach?

Literature review should be extended. Clearly define the gap based on literature review. In the literature review you should discuss why you have used BWM and not for example FUCOM, PIPRECIA, SWARA, LBWA etc method. Some important references published in the last two years are missing. I suggest authors to read and add below references from the field: doi.org/10.31181/dmame2003078m;
doi.org/10.31181/dmame2003019d;
doi.org/10.31181/oresta2001072v;
doi.org/10.31181/oresta1902001k;
doi.org/10.31181/rme200101034c.

You should better organize sensitivity analysis section.

Congratulations to the authros for quality research.

Reviewer #2: The paper is well structured.

The authors should better define motivations for their research.

Why you have used EDAS method?

Why you have used BWM?

Sensitivity analysis should be better organized.

Literature review section should be revised. Add more recent research papers from the filed.

Add more discussion on the results.

Add managerial applications.

Add more future directions in conclusions.

What are limitations of proposed methodology?

6. PLOS authors have the option to publish the peer review history of their article (what does this mean?). If published, this will include your full peer review and any attached files.

Reviewer #1: No

Reviewer #2: No

---

## [Author Response · Author response to Decision Letter 0]

22 Jan 2021

Point to point answers to the reviewers’ comments and suggestions

Reviewer #1: 

Thank you for inviting me as a reviewer for manuscript titled Hybrid BW-EDAS MCDM Methodology for Optimal Industrial Robot Selection; Journal PLOS ONE, Manuscript No: PONE-D-20-36145.

Presented methodology has potential in decision making field and I am giving support to the authors for investigation this topic. The strengths of this paper are: Relevant topic; Flow of the paper; and Explanation of the methods. 

However, the author(s) need to consider the following points as limitation or further scope for refining the paper: 

Thank you for the value able comments and suggestions to improve the manuscript. We have improved the manuscript according to your comments and suggestions. 

Comment # 1: Need to better highlight the novelty of study in the introduction.

Answer: The novelty of the study is included and highlighted according to the suggestions.

Comment # 2: I suggest authors to clearly summarize what specific advantages brings your approach.

Answer: The advantage of this methodology is due to the consistent behavior of BWM, fewer calculation cost of EDAS method and a wide application are of both these methods. That is mentioned and highlighted in the paper. 

Comment # 3: The authors didn’t explain why we need this research? What are motivations (Research Gaps) for this research? Clearly define motivations for your research.

Answer: Robot selection is a very big issue in this era due to large number of manufacturer and vast variety of robots in the market. So, there should be a user friendly and easy to understand methodology that can be adopted for this selection. The existing methods can have non consistent behavior or huge calculation cost that is rectified using this methodology. It is included and highlighted in the manuscript.

Comment # 4: What are limitations of existing approaches in the literature and can be eliminated by this study?

What are benefits of this approach?

Answer: Limitations of existing approaches are the non-consistency of pairwise comparisons for example in AHP comparisons can be non-consistent and the calculation cost of fuzzy ranking methods, for example triangular TOPSIS method. The proposed approach have following benefits, 1) weight deriving process is consistent, 2) it required fewer pairwise comparisons, 3) user friendly for DMs to provide opinions, 4) less calculation cost. It is included and highlighted in the paper.

Comment # 5: Literature review should be extended. Clearly define the gap based on literature review. In the literature review you should discuss why you have used BWM and not for example FUCOM, PIPRECIA, SWARA, LBWA etc. method. Some important references published in the last two years are missing. I suggest authors to read and add below references from the field: 

doi.org/10.31181/dmame2003078m;

doi.org/10.31181/dmame2003019d;

doi.org/10.31181/oresta2001072v;

doi.org/10.31181/oresta1902001k;

doi.org/10.31181/rme200101034c.

Answer: Study gap is defined in introduction section and provided the benefits of use of BWM and reasons for not using other methods for example FUCOM, PIPRECIA, SWARA, LBWA etc. method. Important references are cited in the literature. It is included and highlighted in the paper.

Comment # 6: You should better organize sensitivity analysis section.

Answer: The sensitivity analysis is organized by improving the analysis and included in the paper.

Reviewer #2: 

The paper is well structured.

Thank you for the value able comments and suggestions to improve the manuscript. We have improved the manuscript according to your comments and suggestions. 

Comment # 1: The authors should better define motivations for their research.

Answer: Motivation of the paper is better defined and highlighted in the paper.

Comment # 2: Why you have used EDAS method?

Answer: EDAS method is used due to its simplicity, less calculation cost and huge application area. 

Comment # 3: Why you have used BWM?

Answer: This methodology have following advantages, 1) weight deriving process is consistent, 2) it required fewer pairwise comparisons, 3) user friendly for DMs to provide opinions.

Comment # 4: Sensitivity analysis should be better organized.

Answer: The sensitivity analysis is organized by improving the analysis and included in the paper.

Comment # 5: Literature review section should be revised. Add more recent research papers from the filed.

Add more discussion on the results.

Answer: More discussion in introduction section is added and highlighted.

Comment # 6: Add managerial applications.

Answer: Managerial applications are included in paper as a section and highlighted.

Comment # 7: Add more future directions in conclusions.

Answer: Some future directions are added and highlighted.

Comment # 8: What are limitations of proposed methodology?

Answer: This method do not provide the process of suitable criteria selection among large number of criteria for the specific requirement. This method cannot handle fuzzy information.

---

## [Decision Letter · Decision Letter 1]

26 Jan 2021

Hybrid BW-EDAS MCDM Methodology for Optimal Industrial Robot Selection

PONE-D-20-36145R1

Dear Dr. Chu,

We’re pleased to inform you that your manuscript has been judged scientifically suitable for publication and will be formally accepted for publication once it meets all outstanding technical requirements.

Kind regards,

Dragan Pamucar

Academic Editor

PLOS ONE

Additional Editor Comments (optional):

Reviewers' comments:

Reviewer's Responses to Questions

**Comments to the Author**

1. If the authors have adequately addressed your comments raised in a previous round of review and you feel that this manuscript is now acceptable for publication, you may indicate that here to bypass the “Comments to the Author” section, enter your conflict of interest statement in the “Confidential to Editor” section, and submit your "Accept" recommendation.

Reviewer #1: All comments have been addressed

Reviewer #2: All comments have been addressed

2. Is the manuscript technically sound, and do the data support the conclusions?

Reviewer #1: Yes

Reviewer #2: Yes

3. Has the statistical analysis been performed appropriately and rigorously? 

Reviewer #1: Yes

Reviewer #2: Yes

4. Have the authors made all data underlying the findings in their manuscript fully available?

Reviewer #1: Yes

Reviewer #2: Yes

5. Is the manuscript presented in an intelligible fashion and written in standard English?

Reviewer #1: Yes

Reviewer #2: Yes

6. Review Comments to the Author

Reviewer #1: All the reviewers' comments have been addressed carefully and sufficiently, the revisions are rational from my point of view, I think the current version of the paper can be accepted.

Reviewer #2: I am very happy that the authors have addressed the point of my concern by point precisely. No further suggestions come from my side. Therefore, I would like to recommend this manuscript to be published.

7. PLOS authors have the option to publish the peer review history of their article (what does this mean?). If published, this will include your full peer review and any attached files.

Reviewer #1: No

Reviewer #2: No

---

## [Editor Report · Acceptance letter]

29 Jan 2021

PONE-D-20-36145R1 

Hybrid BW-EDAS MCDM methodology for optimal industrial robot selection 

Dear Dr. Chu:

I'm pleased to inform you that your manuscript has been deemed suitable for publication in PLOS ONE. Congratulations! Your manuscript is now with our production department. 

Kind regards, 

on behalf of

Dr. Dragan Pamucar 

Academic Editor

PLOS ONE